# The Correlation between the MBTI-Based Personality Analysis of Anime Characters and Their Popularity

**DOI:** 10.3390/bs14070522

**Published:** 2024-06-22

**Authors:** Ruiyang Tang, Penghao Yang, Ryoga Miyauchi, Yuki Inoue

**Affiliations:** 1Department of Innovation Science, Tokyo Institute of Technology, 3-3-6 Shibaura, Minato-ku, Tokyo 108-0023, Japan; 2Graduate School of Humanities and Social Sciences, Hiroshima University, 1-1-89 Higashisendamachi, Naka-ku, Hiroshima 730-0053, Japan; yangpenghao0@126.com (P.Y.); miyauchi.hifuku@gmail.com (R.M.); 3Department of Industrial Engineering and Economics, Tokyo Institute of Technology, W9-86, 2-12-1 Ookayama, Meguro-ku, Tokyo 152-8552, Japan

**Keywords:** animation, Myers–Briggs Type Indicator (MBTI), character personality, popularity

## Abstract

Anime has become a global phenomenon due to its diverse cultural representations, relatable characters, and unique storytelling. However, there is limited research on the relationship between character personality and popularity. The aim of this study is to understand the relevance of the characters’ personalities to the audience’s evaluation of various characters. This study analyzed the correlation between the MBTI personality evaluations of characters in anime and their popularity, based on the data of the personality of each animation character reviewed by audiences. In this study, 885 characters from 200 anime aimed at a male audience were selected as a sample for research. The results showed that personality traits such as introversion, intuition, and thinking had an impact on the popularity of female characters but not male characters. The overall results were influenced by the larger sample size of female characters. By addressing this question, the study can contribute to the design of a character’s personality and overall success in anime.

## 1. Introduction

As globalization continues to advance, the prevalence of anime that incorporates various cultures has increased [1]. In recent years, “cool” has been an important factor in the widespread recognition of Japanese animation around the world, which has generated interest and love among audiences in many countries [2]. The setting, story, and characters in anime can influence the audience [3]. Studies on anime characters reveal that main characters can often be infused with emotions such as love, happiness, and a sense of existence, which allows the audience to establish a connection with them [4]. Character behavior is an important way to portray characters and can also influence viewers, especially children [5,6].

To this day, it remains unclear what kind of character personality the audience likes and how it affects the popularity of the character. As anime is a visual art, previous studies mainly focus on visual design and story. Sometimes, character design is based on artistic design strategies, which helps the audience form cognitive impressions of anime characters [7,8,9]. Particularly, as a form of art that makes a direct impact on the eyes, cultural features can be expressed through characters [10,11]. On the other hand, male characters are often portrayed as having broad shoulders and a strong upper body, while in romance, males are often portrayed as gentle and beautiful [12,13]. Additionally, in the story, there are often stereotypical phenomena. Stereotyping is a fixed idea about a particular group of features [14,15].

To create impressive characters that leave an impact on the audience, there are two approaches: one is to focus on the outward appearance and create beautiful characters, enabling the audience to distinguish the characters according to the difference in appearance, and this appearance allows the audience to distinguish the characters [16,17]. The other is to emphasize the internal personality and create characters with traits that are easily accepted by the audience [17]. Personality traits are fundamental characteristics that determine individual behavior [18]. They are psychological structures that enable behavioral tendencies to exhibit traits such as persistence, stability, and consistency, reflecting the consistency and regularity of individual behavior [19]. This form leaves a profound impression on the audience.

In the current society, the Myers–Briggs Type Indicator (MBTI) is a commonly used in general public. According to Myers et al. (1995), MBTI is a diagnostic test that shows an individual’s psychological preferences for perception and decision-making [20]. It categorizes individuals into four dimensions: extraversion (E) vs. introversion (I), sensing (S) vs. intuition (N), thinking (T) vs. feeling (F), and judging (J) vs. perceiving (P). Currently, many audiences also use MBTI to identify the personalities of their favorite characters.

In the research conducted so far, little attention has been paid to the study of character personality, rather than their physical appearance. In studies related to character personality, research has also been conducted on the relationship between personality and other attributes of the character, such as appearance and voice [21]. Furthermore, based on the research of Chory-Asad and Cicchirillo (2005), it has been recognized that viewers tend to seek emotional identification with characters, which is caused by phenomena such as similarity identification [22]. Similarity identification refers to the phenomenon where viewers seek resonance with a character by finding similarities, and the characteristics of resonance that are generated differ depending on the viewer. In order to clarify the characteristics that viewers can resonate with, it is necessary to focus on the attributes of the character. This study focuses on the personality attributes of the character. Specifically, we are interested in how the characters’ personalities relate to the audience’s evaluation of the characters. We propose the following research question:


*RQ: What is the influence of the character’s personality on the viewer’s evaluation of the character?*


By exploring this question, we can understand the relationship between characters’ personalities and the audience’s evaluation of various characters and aid in designing more compelling characters. Regarding personality design, by being aware that characters with easily acceptable personalities can leave a strong impression, this study utilizes data on animated character personalities quantified by the Myers–Briggs Type Indicator (MBTI) for statistical analysis. Therefore, the purpose of this study is to clarify the relationship between the character’s personality and the character’s popularity.

## 2. Previous Research and Hypothesis

In this study, we introduce the influence of anime and cultural products, as well as characters related to this study and the definition and explanation of MBTI, and propose a hypothesis.

### 2.1. Anime and Cultural Products

The anime industry is an important part of the cultural and creative industry and has formed a huge consumer market around the world, becoming a major industry in advanced countries [23]. In the past 20 years, the global acceptance of Japanese culture as an export has attracted widespread academic attention [24]. Anime has already become part of Japan’s soft power in culture [24]. Through economic and cultural aspects, the influence of anime has spread worldwide.

Anime in particular can influence viewers through its setting, story, and characters, and viewers tend to behave and think like the characters in the story [3]. Similarity to characters is important for viewers, and they feel a stronger emotional sense of unity with characters they can empathize with [25]. This means that characters play an important role in anime. Generally, when forming impressions of others, people prioritize morality and personality as more important factors [26]. Viewers also perceive the character’s personality through their actions and the stories they appear in [27]. Khalis and Mustaffa (2017) proposed that character personality is also important in anime, apart from good stories or beautiful characters [17].

In Japanese anime, young girl characters are often seen [28]. According to research on the magical girl anime from the end of the twentieth century by Saito (2014), girl characters possess the characteristics of youthfulness and cuteness, which are related to the desire to temporarily escape from social roles such as wives or mothers, social responsibilities such as work or child-rearing, and to become a new self [29]. Additionally, cuteness is characterized by innocence, blushing easily, and being prone to tears [30]. Despite possessing these cute characteristics, characters who are heroines make decisions, act boldly, protect others, and emphasize themselves [30]. Furthermore, Martinez (2015) points out that these characters not only represent Japanese young girls but are also expressions of otaku desires [31]. For example, in battle anime from the end of the twentieth century, female characters are used to convey the message of overcoming challenges without fear.

### 2.2. About MBTI

#### 2.2.1. Definition of MBTI

The Myers–Briggs Type Indicator (MBTI) is a diagnostic test that shows the psychological preferences of individuals for perception and decision-making [20]. Specifically, it categorizes individuals into four dimensions: extraversion (E) vs. introversion (I) for the direction of interest, sensing (S) vs. intuition (N) for the method of perception, thinking (T) vs. feeling (F) for the method of judgment, and judging (J) vs. perceiving (P) for attitude towards the outer world [20]. According to this theory, the changes in human behavior that may seem accidental actually have consistency and are based on fundamental differences in individual perception and judgment [20].

For the MBTI, four indicators are used to measure an individual’s psychological preferences. Firstly, extraversion (E) vs. introversion (I) refers to whether an individual prefers to focus their energy on the outer world of people and things or the inner world of ideas and images [20]. Those with a high score of “E” are seen as sociable and enjoy participating in various activities and socializing with others [32]. Introverted and “I” personality types focus on their inner world and self-awareness and prefer solitude [33].

Secondly, sensing (S) vs. intuition (N) refers to which information source an individual values more and whether they focus on information received through the five senses or on possibilities presented by the received information [20]. High “S” scorers use existing reality, facts, and past experiences to solve problems [32]. People with high intuition type “N” prioritize new possibilities and the future over experience and the past [34].

Thirdly, thinking (T) vs. feeling (F) refers to the method of decision-making and whether to prioritize objective principles or personal matters and relationships [20]. Those with a high score of “T” tend to seek logical explanations and solutions for most things and are not easily influenced by personal or other people’s wishes during decision-making. On the other hand, individuals who score high in “F” tend to make decisions that fit the situation and people’s perspectives, placing more importance on pleasing others [35].

Fourthly, judging (J) vs. perceiving (P), i.e., the attitude towards the external world, refers to the type of lifestyle preferred and whether one prefers a structured lifestyle or a flexible one [20]. Individuals who score high in “J” tend to prefer a planned and well-prepared lifestyle and seek to control their life and work as much as possible [32]. On the other hand, individuals who score high in “P” tend to prefer a flexible way of life and are more receptive to new experiences and information.

Table 1 shows the 16 types of personalities in the MBTI and their corresponding characteristics.

#### 2.2.2. Relationship between MBTI and Big Five

MBTI is a diagnostic test in psychology, but in previous research in psychology, the Big Five model of personality theory is more commonly used. The reason for this is that MBTI only has four indicators, and its diagnostic range is narrower than the five indicators of the Big Five [46].

The Big Five is a model of personality that defines personality traits and is commonly used as a framework to explain personality. The five elements are defined as openness, conscientiousness, extraversion, agreeableness, and neuroticism [47,48]. According to research by McCrae and Costa (1989), there are several positive relationships between MBTI elements and Big Five elements [49]. Specifically, there are positive relationships between extraversion and the E indicator, openness and the N indicator, agreeableness and the F indicator, and conscientiousness and the J indicator. The relationship between neuroticism and any MBTI elements was not shown, so we will only discuss the other four elements.

Openness is an evaluation of people’s emotions, imagination, curiosity, and various experiences [50]. People with high openness are curious and creative, prefer abstract thinking, and have a wide range of interests [51]. Conscientiousness is an evaluation of behavior, goal achievement, and more [52]. People with high conscientiousness are reliable and sophisticated and tend to act with caution [53]. Extraversion is an evaluation of the number and density of interpersonal relationships, the need for stimulation, and more [47,52]. People with high extraversion enjoy contact with others, are passionate and full of energy, have a constantly positive attitude, and pursue stimulation [54]. Agreeableness is an attitude that exists in a broad range of interpersonal relationships and includes positive attitudes such as intimacy, compassion, trust in others, and kindness, as well as negative attitudes such as anger, suspicion, and cruelty [52].

### 2.3. Hypotheses Building

In real life, extraverted individuals are perceived to be more likely to feel happy and satisfied with their lives [55]. They are also more likely to influence others and make those around them feel more enjoyable. As such, extraverted individuals are considered to be more attractive [56]. Feiler and Kleinbaum (2015) found that extraversion has two important effects on interpersonal relationships, one of which is the popularity effect, meaning that extraverts have more friends than introverts [57]. Van et al. (2010) also found that highly extraverted adolescents are well-liked by others [58]. Based on these findings, it can be inferred that high levels of extraversion (E) lead to higher individual evaluations regardless of gender. Therefore, we propose the following hypothesis:

**H1.** 
*The higher the level of the extroversion dimension (E) of a particular anime character, the more popular the character will be.*


According to the study by Chory-Assad and Cicchirillo (2005), viewers empathize with characters who they perceive as similar to themselves and seek emotional connection [22]. Therefore, it is believed that viewers tend to prefer characters who have personalities similar to their own. Additionally, Yamamoto, Lembright, and Corrigan (1966) found that popularity is related to both creativity and intelligence, with creativity being more strongly associated with boys and intelligence being more strongly associated with girls [59]. McCrae and Costa (1989) found a positive relationship between the MBTI’s intuition dimension and the Big Five’s openness dimension [49]. Based on these findings, it is speculated that higher levels of intuition (N) would lead to higher individual evaluations, regardless of gender. Therefore, we propose the following hypothesis:

**H2.** 
*The higher the level of the intuition dimension (N) of a particular anime character, the more popular the character will be.*


In real life, it is believed that individuals who have a high level of cooperativeness are kind to others [52] and are evaluated positively. Van et al. (2010) found that high levels of cooperativeness are positively related to popularity [58]. Stavrova, Evans, and van Beest (2022) found in their study on trust in interpersonal relationships that individuals with high levels of cooperativeness are perceived as thoughtful and trustworthy and are more likely to be chosen as group members or partners [60]. Additionally, McCrae and Costa (1989) found a positive correlation between cooperativeness and the feeling function of the MBTI [49]. Based on these findings, it is speculated that individuals with a high level of the feeling function (F) will be evaluated positively regardless of gender. Therefore, we propose the following hypothesis:

**H3.** 
*The higher the level of the feeling dimension (F) of a particular anime character, the more popular the character will be.*


In addition, according to Mervielde and De Fruyt (2000), child popularity is related to three indicators when evaluated using the Big Five, and one of them is related to honesty [61]. Furthermore, Frankowski et al. (2016) found that individuals with low levels of honesty perceive themselves as relatively unpopular [62]. Individuals with high levels of honesty are thought to engage in positive behaviors and be positively evaluated by others [53]. McCrae and Costa (1989) also found a positive relationship between honesty as measured by the Big Five and the judgment dimension in the MBTI [49]. Based on these findings, it is hypothesized that individuals with high levels of judgment dimension (J) will be evaluated more positively regardless of gender. Therefore, we propose the following hypothesis:

**H4.** 
*The higher the level of the judgment dimension (J) of a particular anime character, the more popular the character will be.*


## 3. Method

### 3.1. Data Resource

About the data source, this study selected 200 broadcasted anime as of February 2022. The selection criteria for the anime were based on the popularity ranking of romantic anime on the website “Minna no Ranking” (https://ranking.net/rankings/best-lovecomedy-animes, accessed on 13 June 2022) and the popularity vote for anime characters on the website “Uniten” (https://www.uniten.net/, accessed on 13 June 2022), which had already been posted and had a vote count of 50 or more.

Regarding the works, only television anime videos were selected. In cases where the same series and major characters were the same, they were regarded as one anime. If the same series had different major characters, they were considered multiple anime. Since male-oriented anime was predominant in this database, to avoid bias in the analysis, clearly female-oriented anime was excluded, and a total of 137 anime were selected as research subjects. However, considering that 137 anime might be insufficient as a sample size for statistical analysis, additional selections were made to reach 200 anime. Following the popularity ranking of “Popular Anime List” published on the website “Anikore” (https://www.anikore.jp/pop_ranking/, accessed on 13 June 2022), 63 male-oriented anime other than the above 137 were selected. Thus, a total of 200 anime were selected as subjects for analysis.

Then, the top 5 characters in the character ranking by vote rate for each anime were selected as research subjects from the website “Uniten” (https://www.uniten.net/, accessed on 13 June 2022). However, multiple characters may enter the same rank, and even if they are ranked in the top 5 in one anime, there may be 6 characters included. Therefore, a total of 1017 characters were selected as research subjects. However, when checking the histogram of the vote rate, it was confirmed that some samples with excessively few votes become outliers. As a result, samples with a vote rate of less than 2% were deleted, leaving a total of 996 characters. This is explained in detail in the section on the dependent variable.

Regarding the scoring of characters by MBTI, data were obtained using the website “Personality Database” (https://www.personality-database.com/, accessed on 13 June 2022), where registered users can evaluate characters with MBTI. However, if there were too few evaluators, the data were excluded from the analysis. Therefore, we narrowed down to characters with 5 or more evaluators (including 5), resulting in a total of 885 characters as the subject of analysis. The breakdown consists of 267 male characters (30%) and 618 female characters (70%).

### 3.2. Dependent Variable

The study aims to explore the relationship between personality traits and the popularity of each character. Directly measuring the popularity of characters is difficult, so the percentage of votes received by each anime character in a viewer poll was considered as a proxy variable for popularity. In this study, the selected dependent variable was the percentage of votes received by each character (measured in the range of 0–100%). Upon checking the histogram, it was confirmed that the distribution of data was skewed to the right. Therefore, the natural logarithm of the percentage of votes received by each character was taken. Upon checking the distribution with a QQ plot, some outliers were identified. Finally, it was confirmed that removing samples with less than 2% of votes would resolve the skewness in the distribution. Thus, the final dependent variable in this study was the natural logarithm of the percentage of votes received by the selected characters. In the analysis model, it is denoted as “$y_{i}^{Votes}$”.

### 3.3. Independent Variables

The Myers–Briggs Type Indicator (MBTI) was used to quantify the personality of each character. In this study, the degrees of four indicators of MBTI for each character—“extraversion (E)”, “intuition (N)”, “feeling (F)”, and “judging (J)”—were set as independent variables. The independent variables in this study are the degrees of the four MBTI indicators for each character, which range from 0 to 1. In the analysis model, they are denoted as “$x_{i}^{E}$”, “$x_{i}^{N}$”, “$x_{i}^{F}$”, and “$x_{i}^{J}$”.

### 3.4. Control Variables

#### 3.4.1. Gender of Characters

The gender of characters has the potential to significantly influence voting rates, as the gender of the characters that are popular may vary across anime. For instance, in anime targeted towards males, female characters may be more prevalent, which could result in a higher proportion of female characters receiving high votes in character rankings. Therefore, it is necessary to control the influence of the gender of characters. This variable is a dummy variable, with males being coded as 1 and females as 0. In the analysis model, this variable is denoted as “$x_{i}^{Gender}$”. This variable was not used in gender-specific analyses.

#### 3.4.2. Role of Characters

The role of characters is to give impressions and convey information to viewers. Depending on their role, characters have the potential to make different impressions on viewers, which could affect their voting rates. The roles of characters are divided into four categories: male protagonist, female protagonist, other important characters, and supporting characters. These categories are based on the content of “Story Structure: Theory and Text Analysis” by Stanzel (1979) [63]. This variable is a dummy variable, with 1 assigned if the character meets the requirements of the specific role and 0 otherwise. In this study’s model, these variables are denoted as “$x_{i}^{Hero}$”, “$x_{i}^{Heroine}$”, “$x_{i}^{Other.main}$”, and “$x_{i}^{Sub}$”. However, including all dummy variables could result in multicollinearity issues in the analysis. Therefore, the dummy variable “sub” was not included as a reference variable.

When analyzing male characters separately, the control variable “Heroine” was not set as a variable under “Character’s role” since there are no female characters, and only “Hero” and “Other.main” were set as control variables. Similarly, when analyzing female characters separately, only “Heroine” and “Other.main” were set as control variables since there are no male characters.

### 3.5. Empirical Specifications

In this study, linear multiple regression analysis was selected. The reason for choosing this method was to investigate the correlation between MBTI personalities and a character’s popularity. Linear multiple regression analysis helps us understand how different personality traits collectively relate to a character’s popularity. By analyzing the regression coefficients, we can measure the correlation between each independent variable and the dependent variable to identify the personality traits that most significantly correlate with character popularity. Finally, linear multiple regression analysis can control for the effects of other potential confounding variables to draw more accurate conclusions.

In this study, two analyses were conducted. The purpose of Analysis 1 is to explore the correlation between the four MBTI indicators of each character and their popularity. In Analysis 1, the dependent variable was the natural logarithm of the vote percentage for each character included in the analysis, and the four indicators of MBTI for each character were used as the independent variables in a multiple regression analysis. The analysis was conducted for all anime characters included in the study. The purpose of Analysis 2 is to explore how the personality types of characters correlate with their popularity based on gender. In Analysis 2, the same procedure as in Analysis 1 was followed, but the analysis was conducted separately for male and female characters.

For the overall analysis of the characters, the specific analysis model is presented as follows. As previously mentioned, the dependent variable “$y_{i}^{Votes}$” in the linear multiple regression analysis formula is the “natural logarithm of the vote percentage for the selected character”, and the independent variables are the four MBTI indicators for each character: “$x_{i}^{E}$”, $“x_{i}^{N}$”, “$x_{i}^{F}$”, and “$x_{i}^{J}$”. The control variables are dummy variables representing “$x_{i}^{Gender}$”, and the other control variables are dummy variables representing “$x_{i}^{Hero}$”, “$x_{i}^{Heroine}$”, and “$x_{i}^{Other.main}$” in the “Character Role” category. The partial regression coefficients, represented by $\beta_{1}$~$\beta_{8}$, correspond to each variable, and C and $\varepsilon_{i}$ represent the intercept and error terms in the equation. A simple linear least squares multiple regression analysis was used. In the gender-specific analysis, some of the control variables were removed as previously mentioned, and the stepwise method was used to eliminate variables with low importance during the analysis.
$$y_{i}^{Votes}=\beta_{1}x_{i}^{E}+{\beta_{2}x}_{i}^{N}+{\beta_{3}x}_{i}^{F}+{\beta_{4}x}_{i}^{J}+{\beta_{5}x}_{i}^{Gender}+{\beta_{6}x}_{i}^{Hero}+{\beta_{7}x}_{i}^{Heroine}+{\beta_{8}x}_{i}^{Other.main}$$

The text discusses possible problems during the analysis process and their solutions. Multicollinearity was not detected as all VIF (the maximum VIF = 2.084, and the average = 1.481) values were below 4. Non-normality of residuals was not confirmed as the histogram showed a bell-shaped distribution and the QQ plot showed an almost linear shape. The autocorrelation problem was also not confirmed with *p*-values above 0.05 for all characters, male characters only, and female characters only. The heteroscedasticity problem was confirmed in the analysis of all characters and male characters only with *p*-values less than 0.05 but corrected using the Newey–West method to address it.

## 4. Results

### 4.1. Analysis Results for All Characters

Table 2 shows the correlation analysis results between the four MBTI indicators and the voting rates of anime characters. According to the analysis results, a negative significance was observed from the extraversion type (E) (*p* < 0.05) on the vote rate of characters, while a positive significance was observed from the intuition type (N) (*p* < 0.05). In other words, for anime characters as a whole, the higher the intuition type, the higher the vote rate and popularity, while the lower the extraversion and feeling types, the higher the vote rate and popularity. Therefore, among the four hypotheses related to all characters, only hypothesis 2, “The higher the intuition type (N) of an anime character, the more popular the character will be”, was supported by this analysis. The other hypotheses were not supported by this analysis.

### 4.2. Analysis Results for Male Characters

Table 3 shows the correlation analysis results between the four MBTI indicators and the popularity of male anime characters. In the analyzed results, no significance was found for any personality type. That is, it was suggested that the popularity of male anime characters is not related to their personalities. Therefore, none of the four hypotheses were supported for male characters.

### 4.3. Analysis Results for Female Characters

Table 4 shows the correlation analysis results between the four MBTI indicators and the popularity of female anime characters. According to the analysis results, negative significance was found from extraversion (E) (*p* < 0.05) and feeling (F) (*p* < 0.05), while positive significance was found from intuition (N) (*p* < 0.05) on the popularity of characters. That is, for female anime characters, the stronger their intuition is and the weaker their extraversion and feeling are, the higher their popularity and the more popular they become. Therefore, only hypothesis 2 was supported in the four hypotheses related to female characters, and the other hypotheses were not supported.

## 5. Discussion

### 5.1. Interpretation of Results

Based on audience evaluation data regarding the MBTI personality type of anime characters aimed at a male audience, this study investigated the relationship between the MBTI personality indicators of anime characters and their popularity. It aimed to understand how the four MBTI indicators relate to the overall voting rates of anime characters, including male and female characters. The overall analysis and the analysis of female characters yielded similar results, whereas the results for male characters were not significant across all explanatory variables. Furthermore, since the sample size for female characters was larger than that for males, it is inferred that the overall results have a certain relationship with the characteristics of female characters. Therefore, the overall results are not further discussed; instead, the results for males and females are considered separately. The results show that for female characters, factors related to external interaction have no correlation with their popularity, but those with strong introverted, intuitive, and thinking traits are more likely to be popular. These analytical results are summarized in Table 5.

Our findings indicate that the MBTI personality traits of introversion (I), intuition (N), and thinking (T) are significantly correlated with the popularity of female characters, but not with male characters aimed at a male audience. Hoffner and Buchanan (2005) discovered that viewers emotionally connect with characters who share similar personalities [25]. Through such a connection, a cognitive resonance was created between the audience and the characters in the anime. Cognitive resonance is a crucial mechanism that helps viewers feel familiar with characters [64]. Familiarity plays a crucial role in resonance, as viewers tend to feel closer to characters with similar traits [25]. For anime aimed at a male audience, the vast majority of audiences are male. [65]. From this perspective, many otaku (anime fans) exhibit traits like introversion and intuition [66,67], and the link between INT traits and the otaku demographic helps explain the popularity of these female characters.

In this study, there is no correlation between the popularity of male characters and their MBTI indicators, including extraversion or introversion, sensing or intuition, thinking or feeling, judging or perceiving. Several factors may explain why male characters lack resonance. First, the popularity of male characters may be related to factors beyond personality, such as appearance and character positioning. The appearance and leadership role of a male character may be correlated with popularity. Second, according to Reysen et al. (2020), audiences often feel more sexually attracted to characters of the opposite sex [65]. However, for anime aimed at a male audience, the vast majority of audiences are male. Audiences tend to feel greater sexual attraction toward characters of the opposite sex [68]. This sexual attraction may play a crucial role in establishing strong connections and resonance with anime characters. Therefore, for anime primarily aimed at a male audience, the lack of sexual attraction to male characters may hinder male viewers from resonating with them. This can explain the lack of correlation between the personality traits of male characters and their popularity among male viewers.

Our study also uncovers new findings. Previous research (e.g., [29]) has shown a correlation between cuteness, youthfulness, and the popularity of female characters. In our study, personality traits show a significant correlation with popularity. This suggests that the correlation between personality design and character popularity may be more significant than previously believed. The results indicate that factors related to external interaction do not correlate with the popularity of female characters, regardless of judging or perceiving. Characters with strong tendencies towards introversion, intuition, and thinking are more likely to be popular. According to Jung’s theory, judgment and perception processes are categorized into four functions [69]. Our study discovered a correlation between high levels of introverted thinking and introverted intuition and popularity, indicating their roles in judgment and perception processes [33]. People with introverted thinking focus on analyzing information systematically, emphasizing consistency and structure, whereas individuals with introverted intuition gather deeper information and concentrate on future possibilities [69]. Female characters with these traits are more popular in anime aimed at male audiences.

Finally, the study concluded that the appealing female personality characteristics in anime were partly different from the preferred characteristics in reality, which are E, N, F, and J as discussed in the hypotheses building part. Male anime viewers may have biases towards specific personality types or evaluate real-life women differently from fictional female characters [70]. It is suggested that male-oriented anime viewers with INT personality types may struggle to open up to the opposite sex due to their independent and quiet nature [39]. These audiences may have different preferences for anime characters, as these characters can fill certain voids in their real lives through fantasy. These findings highlight the complex interplay between character design and audience preferences, providing new insights for creators and researchers.

### 5.2. Contributions

This study complements previous research on character creation by focusing on characters in anime aimed at a male audience. The Myers–Briggs Type Indicator (MBTI) was used to quantify character personalities and explore their correlation with character popularity. While viewers tend to prioritize a character’s appearance when evaluating them, the study also suggested that personality is an equally important factor. Similarly, the study’s findings suggested that a character’s personality has an influence on their popularity, especially among opposite genders. The study’s results thus demonstrated the impact of a character’s personality on viewers’ evaluation of them.

The study revealed that female characters with intuitive traits are more popular, which aligns with Sunil’s (2020) research on attractive and strong female characters [71]. Sunil found that these characters exhibit calm and rational behavior, indicative of high-level thinking—a result consistent with our study’s finding that female characters with advanced cognitive abilities are also well-liked. Additionally, our study indicates that personality has a stronger correlation with the evaluation of characters of the opposite sex. These findings broaden the scope of character creation research.

Based on the results of this study, it was confirmed that female characters with strong introverted, intuitive, and thinking traits tend to be popular in male-oriented anime. Specific characters that fit this profile in the analyzed works include Yukino Yukinoshita from “My Teen Romantic Comedy SNAFU”, Kaguya Shinomiya from “Kaguya-sama: Love is War”, and Kurisu Makise from “Steins;Gate”. This study’s findings can contribute to the anime production industry in the following ways.

The study suggests that animation companies can use the personality types identified in the research to design characters in anime aimed at a male audience. The “Introverted-I” type emphasizes inner thoughts, while the “Intuitive-N” type focuses on future-oriented thinking and new possibilities. However, characters with the “Thinking-T” type may prioritize objective principles over empathy. These traits are often portrayed as intelligent geniuses or individualistic heroes who overcome challenges. To make animations popular, it is valuable to create female characters with these traits. Specifically, giving female characters INTJ and INTP personalities is meaningful when creating them. Individuals with an INTJ personality have a strong desire to achieve their goals and seek high standards for themselves and others’ abilities [37]. Those with an INTP personality demand logical explanations for all things that interest them [42]. Incorporating these two personality types can enhance the popularity of animated characters created by animation companies.

### 5.3. Limitations and Future Research

There are several limitations to this study, and future challenges related to them can be identified.

Firstly, the sample of this analysis included 885 characters from 200 anime. However, as mentioned in the sampling section, many of the anime studied in this research contain romantic elements. Due to differences in anime genres, there is a possibility that the preferred character personality types and character evaluations may differ. Therefore, future studies could focus on a wider range of anime genres and types and study more characters for analysis.

Secondly, this study did not examine the social attributes of the audience. According to Yildirim and Aydın (2012) [72], the audience’s social attributes like age, gender, education, and marital status may vary, which could relate to their evaluation of the work and characters. Therefore, future studies could narrow down the social attributes of viewers and conduct research on viewers with specific social attributes.

Thirdly, this study used MBTI to score the personalities of the characters. However, compared to the widely used Big Five personality model, the scope of evaluation is narrower. Therefore, in future studies on anime characters, other psychological scoring methods could be used to evaluate and analyze character personalities more comprehensively.

## 6. Conclusions

Anime’s global popularity is attributed to its unique ability to provide diverse cultural expressions, relatable characters, and compelling storytelling. However, despite its increasing popularity, research on the relationship between character personality and character popularity is limited. This study explored the correlation between character personality and audience perception of characters in anime aimed at a male audience. We utilized the data on animated character personalities quantified by MBTI to explore the relationship between character personality traits and the popularity of 200 anime and 885 characters.

The study concluded that female characters with strong tendencies towards introversion, intuition, and thinking (INT) are more popular. This suggests that cognitive resonance is crucial, as the audience feels a sense of familiarity and connection with characters who share similar traits. Anime fans often exhibit INT traits, which may contribute to the affinity and attractiveness of characters. Conversely, the study found no significant relationship between the personality traits of male characters and their popularity. The popularity of male characters may be related to factors beyond personality and differences in how male and female characters resonate with the audience. The variation implies that visual appeal or the character’s role in the story may play a more significant role in determining the popularity of male characters. It also highlights the significance of examining the relationship between audience demographics and expectations in connection with male characters.

This study has contributed to the theoretical development regarding the relationship between character popularity and personality traits. By exploring the relationship between character personality and popularity, this study fills the gap in understanding how personality traits influence character popularity and expands the theoretical framework for developing anime characters. Female characters with introversion, intuition, and thinking (INT) traits are more popular, emphasizing the significance of cognitive resonance. The findings of this study also have practical implications for designing anime characters. By incorporating popular personality traits, especially those linked to cognitive resonance, creators can boost the success and popularity of their characters. For female characters, emphasizing introversion, intuition, and thinking traits in female characters may enhance their appeal.

The study’s sample was restricted to 885 characters from 200 anime with romantic elements, lacking diversity in genres, and the use of MBTI resulted in a narrow evaluation of character personalities. Future research could broaden the sample to cover various anime genres, explore the impact of viewers’ social attributes (e.g., age, gender, and education level) on character evaluations, and utilize advanced personality assessment tools (e.g., the Big Five personality model) for a thorough analysis of character traits.

In conclusion, this study enhances the theoretical understanding of character popularity in anime and provides practical insights for character design. By understanding the psychological mechanisms behind character preference, creators can craft more engaging and relatable characters, ultimately enriching the viewing experience and boosting the success of anime productions.

## Figures and Tables

**Table 1 behavsci-14-00522-t001:** Personality types based on the MBTI.

Personality Types	Characteristic
ISTJ (Introverted, Sensing, Thinking, Judging)	This type of person is quiet and serious-minded [36].
ISFJ (Introverted, Sensing, Feeling, Judging)	This type of person is friendly and responsible [37].
INFJ (Introverted, Intuitive, Feeling, Judging)	This type of person is perceived to be caring, creative, encouraging, and hardworking [38].
INTJ (Introverted, Intuitive, Thinking, Judging)	This type of person has a strong motivation to achieve their goals [39].
ISTP (Introverted, Sensing, Thinking, Perceiving)	This type of person takes quick action to solve problems and find feasible solutions [40].
ISFP (Introverted, Sensing, Feeling, Perceiving)	This type of person is sensitive and kind” [37].
INFP (Introverted, Intuitive, Feeling, Perceiving)	This type of person is idealistic and loyal to their values [41].
INTP (Introverted, Intuitive, Thinking, Perceiving)	This type of person seeks logical explanations for everything they are interested in [42].
ESTP (Extraverted, Sensing, Thinking, Perceiving)	This type of person is flexible and values immediate results [43].
ESFP (Extraverted, Sensing, Feeling, Perceiving)	This type of person is friendly and accepts everything happily [41].
ENFP (Extraverted, Intuitive, Feeling, Perceiving)	This type of person is imaginative [37].
ENTP (Extraverted, Intuitive, Thinking, Perceiving)	This type of person is interested in new things [37].
ESTJ (Extraverted, Sensing, Thinking, Judging)	This type of person is practical and fact-based [44].
ESFJ (Extraverted, Sensing, Feeling, Judging)	This type of person is serious and friendly [20].
ENFJ (Extraverted, Intuitive, Feeling, Judging)	This type of person is caring and responsible [41].
ENTJ (Extraverted, Intuitive, Thinking, Judging)	This type of person is decisive and can always serve as a leader [45].

**Table 2 behavsci-14-00522-t002:** Results of regression analysis for the vote rate of anime characters.

	Complete Model		Optimal Model	
	Estimate	Standard Error	Estimate	Standard Error
Extraversion (E)	−0.083 *	0.040	−0.086 *	0.040
Intuition (N)	0.111 *	0.045	0.108 *	0.045
Feeling (F)	−0.081 ^†^	0.041	−0.082 *	0.041
Judging (J)	−0.024	0.043		
Gender	−0.149 **	0.050	−0.150 **	0.050
Hero	0.364 ***	0.074	0.364 ***	0.074
Heroine	0.617 ***	0.058	0.618 ***	0.058
Other.main	0.329 ***	0.051	0.329 ***	0.051
Constant	2.391 ***	0.066	2.406 ***	0.060
Adj_R-squared	0.163		0.164	
ΔR-squared	0.001			

Notes: *** denotes *p* < 0.001, ** denotes *p* < 0.01, * denotes *p* < 0.05, and ^†^ denotes *p* < 0.1.

**Table 3 behavsci-14-00522-t003:** Results of regression analysis for male characters.

	Complete Model		Optimal Model	
	Estimate	Standard Error	Estimate	Standard Error
Extraversion (E)	−0.013	0.085		
Intuition (N)	0.065	0.093		
Feeling (F)	−0.039	0.083		
Judging (J)	0.073	0.090		
Hero	0.303 **	0.100	0.308 **	0.098
Other.main	0.225 *	0.099	0.231 *	0.098
Constant	2.268 ***	0.114	2.313 ***	0.081
Adj_R-squared	0.021		0.031	
ΔR-squared	0.001			

Notes: *** denotes *p* < 0.001, ** denotes *p* < 0.01, and * denotes *p* < 0.05.

**Table 4 behavsci-14-00522-t004:** Regression analysis results for female characters.

	Complete Model		Optimal Model	
	Estimate	Standard Error	Estimate	Standard Error
Extraversion (E)	−0.109 *	0.045	−0.110 *	0.045
Intuition (N)	0.119 *	0.053	0.117 *	0.051
Feeling (F)	−0.104 *	0.048	−0.105 *	0.048
Judging (J)	0.009	0.049		
Heroine	0.650 ***	0.060	0.651 ***	0.060
Other.main	0.377 ***	0.058	0.377 ***	0.058
Constant	2.386 ***	0.075	2.393 ***	0.067
Adj_R-squared	0.163		0.188	
ΔR-squared	0.001			

Notes: *** denotes *p* < 0.001; * denotes *p* < 0.05.

**Table 5 behavsci-14-00522-t005:** Summary of the analysis results.

MBTI Element	Extraversion (E)	Intuition (N)	Feeling (F)	Judging (J)
Male Characters	No significant	No significant	No significant	Not significant
Female Characters	characters with a strong introversion type tend to be more popular	characters with a stronger intuition type tend to be more popular	characters with a stronger thinking type tend to be more popular	Not significant

## Data Availability

The data presented in this study are available on request from the corresponding author.

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
