# Peer review of "The Correlation between the MBTI-Based Personality Analysis of Anime Characters and Their Popularity"

_behavsci, 2024, doi:10.3390/bs14070522_

Round 1
Reviewer 1 Report
Comments and Suggestions for Authors
Explain the reasons for using linear multiple regression analysis in this study.
Discuss the research results with the results of previous research which formed the basis of the hypothesis
The conclusion explains the limitations of the study and suggestions for future research
Author Response
Dear Reviewer,
Thank you very much for your review. We have made detailed revisions based on your feedback and believe that these changes will improve the readability of our study.
Please refer to the attachment below.
Once again, thank you for your review.

Reviewer 2 Report
Comments and Suggestions for Authors
Thank you for a well-written and interesting paper. It is well presented and clearly shared.
I only have one comment. The conclusions section is quite weak and does not clearly share the contributions to (i) theory/literature, (ii) practice (completely missing), and (iii) future research. Can you please improve that? Perhaps a paragraph on each?
Author Response

(The authors gave the same response as above.)
